# Association between median household income, state Medicaid expansion status, and COVID-19 outcomes across US counties

Tsikata Apenyo[1], Antonio Elias Vera-Urbina[2], Khansa Ahmad[1,3,4], Tracey H. Taveira[1,3,5], Wen-Chih Wu[1,3,4,5,6] *

1 Division of Biology and Medicine, The Warren Alpert Medical School of Brown University, Brown University, Providence, Rhode Island, United States of America, 2 Department of Biology, The University of Puerto Rico, San Juan, Puerto Rico, United States of America, 3 Department of Internal Medicine, The Providence Veterans Affairs Medical Center, Providence, Rhode Island, United States of America, 4 Department of Internal Medicine, Lifespan Hospitals, Providence, Rhode Island, United States of America, 5 College of Pharmacy, The University of Rhode Island, Kingston, Rhode Island, United States of America, 6 Department of Epidemiology, The School of Public Health at Brown University, Providence, Rhode Island, United States of America

* wen-chih.wu@va.gov

**Data Availability Statement:** Replication data stored in Harvard Dataverse© for public access and replication. The URL for the dataset is: https://doi.org/10.7910/DVN/QXT2KS.

## Abstract

### Objective

To study the relationship between county-level COVID-19 outcomes (incidence and mortality) and county-level median household income and status of Medicaid expansion of US counties.

### Methods

Retrospective analysis of 3142 US counties was conducted to study the relationship between County-level median-household-income and COVID-19 incidence and mortality per 100,000 people in US counties, January-20th-2021 through December-6th-2021. County median-household-income was log-transformed and stratified by quartiles. Multi-level-mixed-effects-generalized-linear-modeling adjusted for county socio-demographic and comorbidities and tested for Medicaid-expansion-times-income-quartile interaction on COVID-19 outcomes.

### Results

There was no significant difference in COVID-19 incidence-rate across counties by income quartiles or by Medicaid expansion status. Conversely, for non-Medicaid-expansion states, counties in the lowest income quartile had a 41% increase in COVID-19 mortality-rate compared to counties in the highest income quartile. Mortality-rate was not related to income in counties from Medicaid-expansion states.

### Conclusions

Median-household-income was not related to COVID-19 incidence-rate but negatively related to COVID-19 mortality-rate in US counties of states without Medicaid-expansion.

**Funding:** Yes, this project is supported by the Department of Veterans Affairs, Health Services Research Award IRP-20-003 (WW, project PI).

**Competing interests:** The authors have declared that no competing interests exist.

**Abbreviations:** US, United States; IRR, Incident Rate Ratio; MRR, Mortality Rate Ratio; 95% CI, 95% Confidence Interval.

# Background

Instituted in the 1965, Medicaid has become the largest provider of health insurance in the United States (US) by providing medical access to people with low income and limited resources [1]. The 2010 Affordable Care Act (ACA) sought to decrease the number of uninsured individuals by expanding Medicaid coverage and modifying individual insurance markets [2, 3, 5], but a 2012 Supreme Court decision overturned the requirement that states adopt Medicaid expansion [4, 5]. By January 2020, 14 states had yet to adopt Medicaid expansion [6]. Studies have found that among states who implemented the ACA, the increased access to care has led to early diagnosis of cancers, diabetes, and depression among other health outcomes [7–10], but the relationship between Medicaid expansion and an infectious disease outbreak is unknown.

In 2019, a novel coronavirus (COVID-19), originating from Wuhan City, Hubei Province, China began spreading at an alarming rate [11]. As COVID-19 progressed in the United States (US), the health toll disproportionally impacted African Americans and communities with high prevalence of poor housing conditions [12–14]. In addition, COVID-19 has already been shown to impact individuals with certain pre-existing health conditions at a greater rate [15]. Both federal and state policymakers looked to Medicaid as a central tool in their response to the national emergency [16]. However, whether differences exist in COVID-19 outcomes between communities of Medicaid and non-Medicaid expansion states remains unknown. Moreover, it would be important to quantify differences in outcomes, if any, on the strata that appeared most impacted by COVID-19, the low-income communities, to help the states balance cost versus benefits. This is important because individuals without health insurance coverage are likely to be more vulnerable to the adverse health outcomes related to COVID-19. In the states that did not implement Medicaid expansion, 30 percent of low-income workers were uninsured before COVID-19 [17]. This number was less than half in Medicaid expansion states [17].

We thereby sought to investigate the impact of COVID-19 in the counties nationwide according to their socio-economic status and investigate whether the impact varies by counties of states with Medicaid expansion versus those without Medicaid expansion. We stratified the US counties by its median household income and compared on their COVID-19 incident and mortality rates. We hypothesized that county-level median household income would be inversely related to COVID-19 incidence and mortality rates. Additionally, we hypothesize that a state's Medicaid expansion status will alter the association between county-level median household income and COVID-19 outcomes.

# Methods

Providence Veterans Affairs Medical Center Institutional Review Board, formal waiver of approval due to non-Human subject research, project 1660196.

We conducted a longitudinal, retrospective analysis of data of the US counties and District of Columbia (n = 3142) using 2010–2019 baseline data from the Centers for Disease Control and Prevention and the US Census Bureau and related them to the COVID-19 outcome data from the John Hopkins Coronavirus Resource Center, 2020 [18–21]. Counties from US territories (American Samoa, Guam, Northern Mariana Island, Puerto Rico and US Virgin Islands, n = 105) were not part of the analysis [18]. All data used in this study were publicly available; therefore, the study met the criteria for exemption by the Providence Veterans Affairs Medical Center Institutional Review Board.

## Main exposure variables

County-level median household income per annum for each county was collected from the 2018 US Census Bureau's Small Area Income and Poverty Estimates (SAIPE) [19], and log-transformed (ln X +1) to approximate normality prior to stratification by quartiles. We defined Medicaid expansion states as those that had adopted expansion efforts as of the first case of COVID-19 in the United States on January 20, 2020 (Listing in S1 Table) [6]. Counties in 36 states plus Washington, DC, were included in the Medicaid expansion group, while counties in 14 states were in the non-Medicaid expansion group (S1 Table).

## Outcome

The main outcomes of our study were the cumulative COVID-19 incidence rate and mortality rate per 100,000 of the population from January 20th to December 6th, 2020 [21]. The cumulative COVID-19 incidence and mortality rates of the respective US counties were obtained from the John Hopkins Coronavirus Resource Center, divided by the county population and reported as incidence and mortality rates per 100,000, respectively.

## Covariates

Data on age and gender were collected using 2010 US Census Bureau data as the elderly and men have been reported as possessing a higher risk of COVID-19 mortality [19]. The COVID-19 pandemic has been shown to afflict minority races in the US to a greater degree; therefore, we included data for racial composition of counties: percentage of White, Black, and Hispanic residents using US Census Bureau data from 2014–2018. Population density (population per square feet of land area) was calculated from the county population from 2010 US census divided by the square foot area of the county to account for overcrowding in a community. In addition to median-household income, we also abstracted data that were confirmatory of the socioeconomic status of the communities such as unemployment rate (2019), percentage of population age >25 years without high school diploma (2014–2018), and percentage of population age <65 years without health insurance (2018) [22]. Access to care was assessed by number of hospitals per county (2017).

Since diabetes mellitus, obesity, and smoking are known risk factors for worse outcomes in COVID-19 [23], we obtained the percentage of the population aged >20 years diagnosed with diabetes mellitus, with obesity, and percentage of adults who are current smokers from the Centers for Disease Control and Prevention from 2016–2018 [20, 23].

## Statistical analysis

Baseline characteristics for the counties were described by mean ± standard deviation (SD) and range for continuous variables and percentage for categorical variables. Counties were stratified by quartiles of log-transformed county-level median household income per annum, which comprised of the following median household income ranges: Q1 ($25,385 - $43,681); Q2 ($43,688 - $50,565); Q3 ($50,568 - $58,838); Q4 ($58,848 - $140,382). Linear regression was used to test for trend of baseline characteristics across the income quartiles.

We used a multilevel mixed-effects generalized linear model with a negative binomial distribution and log link function to study the relationship between quartiles of log-median household income and COVID-19 outcomes across US counties: incidence and mortality, in a separate fashion, using Q4 as the referent. We applied a random intercept for states to account for clustering effect due to similarities in health policy for counties within the same state and specifying an unstructured covariance matrix. Using county population as the offset in the

model, the outcomes reported were incidence rate ratios (IRR) and mortality rate ratios (MRR) of COVID-19 across income quartiles of the counties, respectively. In a stepwise fashion, we first adjusted for demographics age over 65 years old, gender, and race (Model 1); followed by population density, diabetes, obesity, current smoking status, state Medicaid expansion status and number of hospitals (Model 2). The percentage of population without high school diploma under 25 years old and population without health insurance were not included in the model given their significant correlation with the median household income per county ($r$ = -0.36, $P<0.001$) and the Medicaid expansion status ($r$ = -0.63, $P<0.001$) variables, respectively. We tested for interaction between quartiles of log-median household 'income quartile-times-Medicaid expansion status' on COVID-19 outcomes in Model 2. If the interaction was significant, the above analyses were repeated, stratified by counties of states with Medicaid Expansion (n = 1,814) and counties of states without Medicaid Expansion (n = 1,328).

Sensitivity analyses were performed to replace median age in lieu of % over 65 years old. All analyses were performed using STATA/SE version 11.2 software (StataCorp LP, College Station, TX). A 2-sided p-value of < 0.05 was considered significant. Replication data stored in Harvard Dataverse© for public access and replication.

## Results

As of December 6, 2020, there were a total of 14,528,356 COVID-19 cases, and 279,115 COVID-19 deaths, across the 3,142 US counties. The mean (SD) for COVID-19 incidence and mortality were 5,155.01 (4308.24) cases and 88.38 (96.46) deaths per 100,000 population per county, respectively.

The characteristics of the 3,142 counties, overall and stratified by four quartiles of log-transformed median household income were presented in Table 1. Overall, 57.7% of counties were located in Medicaid expansion states. Higher median household income quartiles were associated with higher mean county population and population density, number of hospitals and percentage of white residents. Conversely, lower median household income quartiles were associated with higher percentage of elderly residents (65 years or older), of black or Hispanic population, of unemployed, of population without a high school diploma and of people without health insurance. Counties of lower income quartiles were also associated with a higher prevalence of diabetes, obesity and smoking and had a lower likelihood of belonging to a state that adopted Medicaid-expansion.

The mean number of COVID-19 cases and deaths per 100,000 population across counties from different income quartiles were described in Table 2. Specifically, the cases per 100,000 population attributed to COVID-19 were 5,121.08 ± 2,471.59 for counties in the lowest income quartile as compared to 5,033.77 ± 5,705.18 for counties in the highest income quartile. There was no significant association between COVID-19 incidence and quartiles of household income in unadjusted and adjusted analyses. The only exception was in the fully adjusted model, where counties from income quartile 2 (5,299.32 ± 5338.71 COVID-19 cases per 100,000) had a 10% increase in the risk of COVID-19 incidence compared to counties in the income quartile 4 (IRR 1.10, 95% CI: 1.04–1.17). The interaction between income quartile and Medicaid expansion status' was not significant for COVID-19 incidence (P values 0.07 to 0.20 Q1-3).

Conversely, there was a significant association between COVID-19 mortality and quartiles of household income. Specifically, the deaths per 100,000 population attributed to COVID-19 were 113.32 ± 87.43, for counties in the lowest income quartile as compared to 72.32 ± 112.19, for counties in the highest income quartile. In the fully adjusted model, counties from income

**Table 1. Counties baseline characteristics by log transformed county-level median household income.**

| Variables (2010–2019, Census, CDC) | Overall Mean ± SD (Range) n = 3,142 | Log Transformed County-Level Median Household Income Quartiles | | | | Linear trend P-value |
|---|---|---|---|---|---|---|
| | | Quartile 1 Mean ± SD n = 786 | Quartile 2 Mean ± SD n = 784 | Quartile 3 Mean ± SD n = 785 | Quartile 4 Mean ± SD n = 787 | |
| County-Level Median Household Income, $ (2018) | 52,794.41 ± 13,880.12 (25,385–140,382) | 38,514.62 ± 3,857.38 | 47,217.37 ± 1,953.39 | 54,293.69 ± 2,368.64 | 71,139.70 ± 13,110.30 | <0.0001 |
| Population (2010) | 98,174.98 ± 312,433.81 (82–9,818,605) | 28,645.91 ± 66,314.95 | 54,073.68 ± 127,899.69 | 83,421.66 ± 196,745.95 | 226,261.71 ± 554,409.79 | <0.0001 |
| Population density per square foot (2010) | 216.10 ± 1,231.37 (0.04–47,505.94) | 107.48 ± 909.25 | 120.44 ± 518.46 | 125.91 ± 305.19 | 509.83 ± 2181.17 | <0.0001 |
| Median Age (2010), years | 40.34 ± 5.06 (21.90–62.70) | 40.41 ± 5.12 | 41.17 ± 5.28 | 40.65 ± 5.17 | 39.11 ± 4.38 | <0.0001 |
| Population >65 years (2010), % | 15.88 ± 4.19 (3.47–43.38) | 16.51 ± 3.99 | 17.07 ± 4.10 | 16.42 ± 3.98 | 13.54 ± 3.77 | <0.0001 |
| Male (2010), % | 49.98 ± 2.22 (43.20–72.10) | 49.98 ± 2.95 | 50.11 ± 2.31 | 49.90 ± 1.58 | 49.91 ± 1.79 | 0.2134 |
| White (2014–2018), % | 76.45 ± 20.18 (0.7–100) | 67.03 ± 24.87 | 79.91 ± 17.72 | 81.26 ± 16.29 | 77.60 ± 17.55 | <0.0001 |
| Black (2014–2018), % | 8.87 ± 14.46 (0–87.4) | 17.86 ± 21.53 | 7.04 ± 11.23 | 4.99 ± 8.37 | 5.60 ± 8.23 | <0.0001 |
| Hispanic Latino (2014–2018), % | 9.21 ± 13.79 (0–99) | 9.54 ± 18.09 | 8.29 ± 12.81 | 9.06 ± 12.14 | 9.95 ± 11.01 | 0.0979 |
| Unemployment rate (2019), % | 4.00 ± 1.48 (0.7–19.3) | 4.93 ± 1.71 | 4.10 ± 1.40 | 3.69 ± 1.16 | 3.28 ± 1.05 | <0.0001 |
| Age >25 years without high school diploma, (2014–2018), % | 13.41 ± 6.34 (1.2–66.3) | 19.22 ± 6.03 | 13.60 ± 5.08 | 11.50 ± 4.95 | 9.31 ± 4.45 | <0.0001 |
| Age <65 years without insurance (2018), % | 11.50 ± 5.04 (2.4–32.2) | 14.00 ± 4.89 | 12.37 ± 4.86 | 10.53 ± 4.67 | 9.11 ± 4.31 | <0.0001 |
| Number of Hospitals per county (2017) | 1.46 ± 2.56 (0–79) | 0.86 ± 0.79 | 1.23 ± 1.38 | 1.48 ± 1.83 | 2.26 ± 4.38 | <0.0001 |
| Age-adjusted population with diabetes mellitus, age >20 years (2016), % | 10.38 ± 3.80 (1.5–33) | 12.71 ± 4.34 | 10.84 ± 3.64 | 9.50 ± 2.91 | 8.45 ± 2.67 | <0.0001 |
| Age-adjusted population with obesity, age >20 years (2016), % | 32.76 ± 5.70 (12.3–57.9) | 35.15 ± 5.79 | 33.77 ± 5.20 | 32.38 ± 4.78 | 29.74 ± 5.55 | <0.0001 |
| Population with reported smoking (2017), % | 17.47 ± 3.63 (6–41) | 20.58 ± 3.82 | 17.93 ± 2.79 | 16.46 ± 2.51 | 14.89 ± 2.53 | <0.0001 |
| Number of Counties in states with Medicaid expansion | 1,814 | 321 | 417 | 514 | 562 | N/A |

quartile 1 had a 22% increase in the risk of COVID-19 mortality compared to quartile 4 (MRR 1.22, 95% CI 1.09–1.35). Furthermore, the interaction 'income quartile*Medicaid expansion status' was significant (P values <0.01, Q1-3), for which subgroup analyses by Medicaid expansion status were conducted. The sensitivity analyses replacing % population over 65 years old with median age of the county population did not significantly change the results.

**Table 2. Association of SARS-COV-2 outcomes as of December 6, 2020 with county-level median household income quartiles.**

| SARS-CoV2 Outcomes (As of December 6, 2020) | County-Level Median Household Income Quartiles | | | |
|---|---|---|---|---|
| | Quartile 1 | Quartile 2 | Quartile 3 | Quartile 4 |
| | N = 786 | N = 784 | N = 785 | N = 787 |
| | IRR / MRR | IRR / MRR | IRR / MRR | IRR / MRR |
| | (95% CI) | (95% CI) | (95% CI) | (95% CI) |
| Cases per 100,000 population (mean ± SD) | 5,121.08 ± 2471.59 | 5,299.32 ± 5338.71 | 5,166.40 ± 2666.65 | 5,033.77 ± 5705.18 |
| Model 1 | 0.96 | 1.05 | 0.96 | REFERENT |
| | [0.90–1.02] | [0.99–1.10] | [0.91–1.01] | |
| Model 2* | 1.04 | 1.10 | 1.00 | REFERENT |
| | [0.97–1.12] | [1.04–1.17] | [0.95–1.05] | |
| Deaths per 100,000 population# (mean ± SD) | 113.32 ± 87.43 | 92.21 ± 109.58 | 75.69 ± 62.84 | 72.32 ± 112.19 |
| Model 1 | 1.16 | 1.15 | 0.99 | REFERENT |
| | [1.06–1.26] | [1.07–1.24] | [0.93–1.06] | |
| Model 2* | 1.22 | 1.18 | 1.02 | REFERENT |
| | [1.09–1.35] | [1.08–1.28] | [0.95–1.10] | |

IRR = Incident Rate Ratio; MRR = Mortality Rate Ratio; 95% CI = 95% Confidence Interval

Model 1: % Population > 65 years, % Male, and % White

Model 2: % Population > 65 years, % Male, % White, Population Density, % Obesity, % Smoking, % Diabetes, Number of Hospitals, Medicaid expansion status according to state policy

*Interaction between income quartiles and Medicaid status was significant (p-value ≤ 0.005) for SARS-COV-2 mortality but not for SARS-CoV2 Cases (p-value ≥ 0.073)

#Differences between means were statistically significant (p-value < 0.0000) for SARS-COV-2 Deaths per 100,000 population but not for SARS-COV-2 Cases per 100,000 population (p-value < 0.6693)

The comparison of baseline characteristics between counties in Medicaid and non-Medicaid expansion states were described in Table 3. Counties from states with Medicaid expansion had a higher population density, percentage of white residents, median household income, unemployment rate, number of hospitals; and a lower percentage of population who were Black, Hispanic, without high school diploma, without health insurance, with diabetes, with obesity or reported being a current smoker.

The association between household income quartiles and COVID-19 mortality by state Medicaid expansion status was depicted in Table 4. In Medicaid-expansion states, the deaths per 100,000 population attributed to COVID-19 were 92.31 ± 128.60, for counties in the lowest income quartile as compared to 70.20 ± 138.43, for counties in the highest income quartile. On the other hand, for non-Medicaid-expansion states, the COVID-19 deaths per 100,000 population were 138.78 ± 89.11, for counties in the lowest income quartile as compared to 73.36 ± 53.55, for counties in the highest income quartile. In fully adjusted analyses, median household income quartiles were associated with COVID-19 mortality only in counties within non-Medicaid-expansion states, such that counties in the lowest income quartile had a 41% increase in COVID-19 mortality compared to counties in the highest income quartile (MRR 1.41, 95% CI: 1.25–1.59). Contrarily, there were no significant differences in COVID-19 mortality risk by income quartiles in counties within Medicaid expansion states (Fig 1).

## Discussion

To our knowledge, this is one of the first investigations of the association between median household income with COVID-19 outcomes at the county level, in Medicaid expansion and non-expansion states. We found no significant difference in COVID-19 incidence across

**Table 3. Counties baseline characteristics by Medicaid expansion status of the state.**

| Variables (2010–2020, Census, CDC, Johns Hopkins Coronavirus Resource Center) | Counties within States with Medicaid Expansion n = 1,814 Mean ± SD | Counties within States without Medicaid Expansion n = 1,328 Mean ± SD | P-value |
|---|---|---|---|
| Population (2010) | 114,149.71 ± 367932.35 | 76,354.07 ± 212778.78 | 0.0008 |
| Population density per square foot (2010) | 289.89 ± 1599.52 | 115.30 ± 275.52 | 0.0001 |
| Median Age (2010), years | 40.75 ± 5.10 | 39.77 ± 4.94 | <0.0001 |
| Population > 65 years (2010), % | 15.94 ± 4.14 | 15.79 ± 4.25 | 0.3138 |
| Male (2010), % | 50.03 ± 2.07 | 49.90 ± 2.41 | 0.1078 |
| White (2014–2018), % | 80.92 ± 18.13 | 70.33 ± 21.23 | <0.0001 |
| Black (2014–2018), % | 5.65 ± 10.59 | 13.28 ± 17.56 | <0.0001 |
| Hispanic Latino (2014–2018), % | 7.62 ± 11.17 | 11.38 ± 16.48 | <0.0001 |
| Overall, County-level Median Household Income (2018), $ | 55,512.16 ± 14866.59 | 49,084.12 ± 11411.09 | <0.0001 |
| Quartile 1, County-level Median Household Income (2018), $ | 38,462.38 ± 3943.10 | 38,550.68 ± 3800.93 | 0.7526 |
| Quartile 2, County-level Median Household Income (2018), $ | 47,417.04 ± 1941.98 | 46,990.50 ± 1944.14 | 0.0022 |
| Quartile 3, County-level Median Household Income (2018), $ | 54,335.76 ± 2317.18 | 54,213.89 ± 2465.68 | 0.4935 |
| Quartile 4, County-level Median Household Income (2018), $ | 72,362.99 ± 14049.30 | 68,089.64 ± 9787.66 | <0.0001 |
| Unemployment rate (2019), % | 4.13 ± 1.63 | 3.81 ± 1.24 | <0.0001 |
| Population without high school diploma, aged >25 years (2014–2018), % | 11.80 ± 5.54 | 15.61 ± 6.69 | <0.0001 |
| Population without health insurance, aged <65 years (2018), % | 8.77 ± 3.26 | 15.23 ± 4.64 | <0.0001 |
| Number of Hospitals per county (2017) | 1.58 ± 2.96 | 1.28 ± 1.86 | 0.0010 |
| Age-adjusted population with diabetes mellitus, aged >20 years (2016), % | 9.70 ± 3.38 | 11.30 ± 4.14 | <0.0001 |
| Age-adjusted population with obesity, aged >20 years (2016), % | 32.12 ± 5.70 | 33.63 ± 5.63 | <0.0001 |
| Population with reported smoking (2017), % | 17.09 ± 3.63 | 17.98 ± 3.39 | <0.0001 |

counties by income quartiles and when sub-stratified by Medicaid-expansion status. However, we found a significant difference in COVID-19 mortality by county median household income, such that COVID-19 mortality was significant higher in counties from the lower compared to the highest income quartiles, but only in states that did not adopt Medicaid-expansion, and not significantly different in counties from Medicaid-expansion states.

There is ample evidence to support that socioeconomic status is related to health outcomes. Our group has shown that the percentage of population living in poverty in communities was associated with a higher cardiovascular and heart failure mortality [24]. We also showed that counties with higher percentage of households living in poor housing conditions had significantly higher risk of COVID-19 incidence and mortality [14]. In this study, we showed that COVID-19 infection affected communities of distinct income strata in a similar fashion, but with a higher mortality risk in communities of lower household income. Multiple mechanisms have been posited to explain poor health outcomes in low-income population. It is possible that people in lower-income communities have worse health at baseline, receive care at lower quality hospitals, receive differential care within a hospital due to lack of health insurance or poor health literacy, and/or there is a lack of access to care outside of the hospital due to lack of health insurance [25, 26]. In this study, the mechanisms for a higher COVID-19 mortality associated with lower-income quartiles compared to the highest are likely multi-factorial. At

**Table 4. Subgroup analysis of Medicaid and non-Medicaid SARS-COV-2 mortality rate as of December 6, 2020 with county-level median household income quartiles.**

| SARS-COV-2 mortality rate (As of December 6, 2020) | Log Transformed County-Level Median Household Income Quartiles | | | | | | | |
|---|---|---|---|---|---|---|---|---|
| | Quartile 1 Medicaid N = 454 MRR [95% CI] | Quartile 2 Medicaid N = 453 MRR [95% CI] | Quartile 3 Medicaid N = 452 MRR [95% CI] | Quartile 4 Medicaid N = 455 MRR [95% CI] | Quartile 1 Non-Medicaid N = 332 MRR [95% CI] | Quartile 2 Non-Medicaid N = 332 MRR [95% CI] | Quartile 3 Non-Medicaid N = 332 MRR [95% CI] | Quartile 4 Non-Medicaid N = 332 MRR [95% CI] |
| Deaths per 100,000 (mean ± SD) | 92.31 ± 128.60 | 71.83 ± 73.10 | 77.18 ± 67.47 | 70.20 ± 138.43 | 138.78 ± 89.11 | 104.13 ± 68.60 | 94.69 ± 79.22 | 73.36 ± 53.55 |
| Model 1 | 1.01 | 1.09 | 0.94 | REFERENT | 1.43 | 1.31 | 1.15 | REFERENT |
| | [0.88–1.15] | [0.97–1.21] | [0.85–1.03] | | [1.30–1.57] | [1.19–1.43] | [1.05–1.26] | |
| Model 2 | 1.06 | 1.12 | 0.97 | REFERENT | 1.41 | 1.28 | 1.14 | REFERENT |
| | [0.90–1.26] | [0.98–1.27] | [0.87–1.07] | | [1.25–1.59] | [1.16–1.42] | [1.03–1.25] | |

MRR (95% CI) = Mortality Rate Ratio (95% Confidence Interval)

Model 1: % Population > 65 years, % Male, and % White

Model 2: % Population > 65 years, % Male, % White, Population Density, % Obesity, % Smoking, % Diabetes, Number of Hospitals, Medicaid expansion status according to state policy

*Interaction between income quartiles and Medicaid status was significant (p-value ≤ 0.005) for SARS-COV-2 mortality but not for SARS-CoV2 Cases

the county level, we found a higher prevalence of obesity, diabetes and smoking in lower income communities to support a lower baseline health status of the lower-income

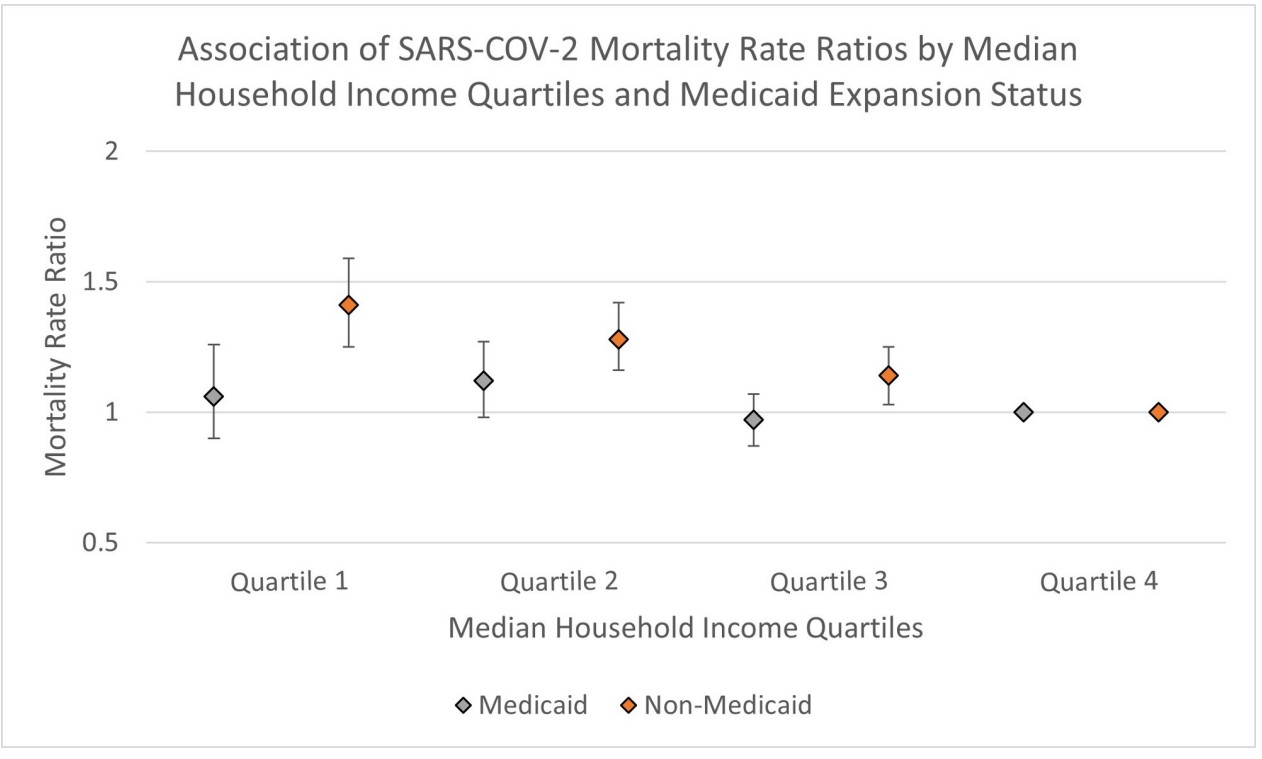

**Fig 1. Association of SARS-COV-2 mortality rate ratios (95% confidence intervals) by median household income quartiles, for counties in Medicaid (gray diamond) and non-Medicaid expansion states (orange diamond), referent = Quartile 4.** In fully adjusted analyses, median household income quartiles were associated with COVID-19 mortality only in counties within non-Medicaid-expansion states (orange diamond), but there were no significant differences in COVID-19 mortality risk by income quartiles in counties within Medicaid expansion states (gray diamond).

communities. We also found a higher prevalence of non-graduation from high school as well as a lack of health insurance in counties within the lowest income-quartiles, which can potentially lead to a lower health literacy and health care access, respectively. All of the above could affect the population behavior including the timeliness towards seeking healthcare when they become ill with COVID-19 as well as post-hospitalization care after discharge.

At the state level, various mechanisms can explain the findings of disparate COVID-19 mortality risk in lower income compared to high income communities, a finding that is significant only in states that did not adopt Medicaid expansion but not significant in Medicaid-expansion states. It is possible that Medicaid expansion is only a marker of the state-level policy towards COVID-19, in terms of mask and social distancing mandates as well as health education and promotion practices for the population, all of which could influence population behavior stated above. In addition, the observed mortality outcome differences across income quartiles in states without Medicaid expansion can also be related to a lower health care access due to lack of insurance after contracting COVID-19, since there was no significant difference in COVID-19 incidence across income quartiles. This is supported by the much higher prevalence of population without health insurance at 14% in the lowest income quartile, compared to 9% in counties from the highest income-quartile (Table 1), a percentage that is twice larger in non-Medicaid expansion states (Table 3).

Over the past decade, studies have shown that in the states that expanded Medicaid coverage, there were improvements in diagnosis, management and mortality of chronic conditions [7–10, 27–29]. Further studies have also investigated the impact on disease mortality rates in Medicaid expansion states on a nationwide scale [30, 31]. In end-stage renal disease, patients had improved 1-year survival rates in Medicaid expansion states [31]. Similarly, a decrease in cardiovascular mortality was observed in states after Medicaid expansion. This was considered to be a benefit of improved access to healthcare for low income individuals by raising the Medicaid eligibility threshold to 138% of the federal poverty level [30, 32]. We believe similar mechanisms may in part explain the differences in COVID-19 mortality at communities of different income strata, especially in non-Medicaid-expansion states. A review of literature shows that individuals without health insurance are less likely to seek health care even when in need [33]. In contrast, it has been shown that when they could afford care, individuals were more likely to utilize healthcare resources [23, 34–36]. Therefore, while a proportion of population in high income communities are able to afford insurance regardless of state Medicaid expansion status, exemplified by similar mortality rates between counties in the highest-income quartiles between Medicaid vs. non-Medicaid expansion states, the highest mortality rate gap is observed in the lowest-income quartiles. Thus, the lack of access to health care is another potential mechanism for COVID-19 mortality disparity in low-income communities from non-Medicaid expansion states.

## Limitations and strengths

The strength of this study is that it is a nationwide study, that utilized cumulative and representative data of US communities in 2020 suitable to assess outcomes as it relates to socio-economic status. Study limitations include its observational design, inability to conclude causality and the potential for residual confounding despite our careful control of known confounders. For example, the use of crude mortality and COVID-19 incidence rates instead of age-adjusted rates, to account for diversity in age distribution in a county, is a limitation and may introduce confounding. As such, we adjusted for age, gender, race and comorbidities of the county population in the final model to minimize the residual confounding. We are aware that policies regarding social distancing and mask mandate may influence the outcomes, it is difficult to

incorporate these into the analyses given the ever-changing nature of these policies through-out the year and the disparate execution of these mandates at the regional level. Instead, we used the cumulative outcome approach to study the Medicaid-expansion policy that was unaltered during 2020. Although some states adopted Medicaid expansion into their state constitution during 2020 (Missouri and Oklahoma), none of them achieved implementation stage during 2020.

## Conclusions and implications

Median-household-income was not related to COVID-19 incidence but negatively related to COVID-19 mortality in US counties of states without Medicaid-expansion. It was unrelated to mortality in counties of states that adopted Medicaid-expansion. Future studies are needed to untangle which state policies have the most impact in the attenuation of the excessive COVID-19 mortality risk associated with socioeconomically disadvantaged communities.

## Supporting information

**S1 Table. List of Medicaid expansion and non-Medicaid expansion states as of January 1, 2020.**
(DOCX)

## Acknowledgments

The views expressed in this paper represent the authors and not the Department of the Veterans Affairs.

## Author Contributions

**Conceptualization:** Tsikata Apenyo, Khansa Ahmad, Tracey H. Taveira, Wen-Chih Wu.

**Data curation:** Tsikata Apenyo, Khansa Ahmad.

**Formal analysis:** Tsikata Apenyo, Antonio Elias Vera-Urbina.

**Investigation:** Tsikata Apenyo, Khansa Ahmad, Tracey H. Taveira, Wen-Chih Wu.

**Methodology:** Tsikata Apenyo, Antonio Elias Vera-Urbina, Khansa Ahmad, Tracey H. Taveira, Wen-Chih Wu.

**Project administration:** Wen-Chih Wu.

**Resources:** Wen-Chih Wu.

**Supervision:** Khansa Ahmad, Tracey H. Taveira, Wen-Chih Wu.

**Validation:** Antonio Elias Vera-Urbina.

**Writing – original draft:** Tsikata Apenyo.

**Writing – review & editing:** Khansa Ahmad, Tracey H. Taveira, Wen-Chih Wu.

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
