## [Editor Report · Decision Letter 0]

11 Jan 2022

PONE-D-21-39999

Association between Median Household Income, State Medicaid Expansion Status, and COVID-19 Outcomes Across US Counties

PLOS ONE

Dear Dr. Wu,

Thank you for submitting your manuscript for review by PLOS ONE. After careful consideration, we have decided that your manuscript does not meet our publication criteria and must therefore be rejected.

As noted, this manuscript is a resubmission of PLOS ONE submission PONE-D-21-03802, which was previously rejected due to concern about the contents of the manuscript. Unfortunately, we do not feel that these concerns have been sufficiently addressed.

In light of the remaining concerns, the manuscript does not meet our publication criteria requiring that conclusions are presented in an appropriate fashion and are supported by the data.

I am sorry that we cannot be more positive on this occasion, but I do hope that you will find these comments useful when deciding how to proceed with your manuscript.

With best wishes,

James Mockridge, PhD

Division Editor

PLOS ONE
---

## [Author Response · Author response to Decision Letter 0]

24 Feb 2022

The response to the reviewers was included as a word file at the end of the uploaded documents

---

## [Decision Letter · Decision Letter 1]

15 Jun 2022

PONE-D-21-39999R1Association between Median Household Income, State Medicaid Expansion Status, and COVID-19 Outcomes Across US CountiesPLOS ONE

Dear Dr. Wu,

Thank you for submitting your manuscript to PLOS ONE. After careful consideration, we feel that it has merit but does not fully meet PLOS ONE’s publication criteria as it currently stands. Therefore, we invite you to submit a revised version of the manuscript that addresses the points raised during the review process.

 Please revise.

We look forward to receiving your revised manuscript.

Kind regards,

Academic Editor

PLOS ONE

Journal Requirements:

Yes, this project is supported by the Department of Veterans Affairs, Health Services Research Award IRP-20-003 (WW, project PI)

Please state what role the funders took in the study. If the funders had no role, please state:

If this statement is not correct you must amend it as needed. Please include this amended Role of Funder statement in your cover letter; we will change the online submission form on your behalf.

3. Thank you for stating the following in the Acknowledgments/ Funding Section of your manuscript:

"This project is supported by the Department of Veterans Affairs, Health Services Research Award IRP-20-003 (WW, Project PI)"

"Yes, this project is supported by the Department of Veterans Affairs, Health Services Research Award IRP-20-003 (WW, project PI)"

4. We noted in your submission details that a portion of your manuscript may have been presented or published elsewhere. Please clarify whether this publication was peer-reviewed and formally published. If this work was previously peer-reviewed and published, in the cover letter please provide the reason that this work does not constitute dual publication and should be included in the current manuscript.

5. Please amend your list of authors on the manuscript to ensure that each author is linked to an affiliation. Authors’ affiliations should reflect the institution where the work was done (if authors moved subsequently, you can also list the new affiliation stating “current affiliation:….” as necessary).

8. Thank you for including your ethics statement on the online submission form:

"Providence Veterans Affairs Medical Center Institutional Review Board, formal waiver of approval due to non-Human subject research, project 1660196."

To help ensure that the wording of your manuscript is suitable for publication, would you please also add this statement at the beginning of the Methods section of your manuscript file.

Additional Editor Comments (if provided):

Reviewers' comments:

Reviewer's Responses to Questions

**Comments to the Author**

1. If the authors have adequately addressed your comments raised in a previous round of review and you feel that this manuscript is now acceptable for publication, you may indicate that here to bypass the “Comments to the Author” section, enter your conflict of interest statement in the “Confidential to Editor” section, and submit your "Accept" recommendation.

Reviewer #1: All comments have been addressed

Reviewer #2: All comments have been addressed

2. Is the manuscript technically sound, and do the data support the conclusions?

Reviewer #1: Yes

Reviewer #2: Yes

3. Has the statistical analysis been performed appropriately and rigorously? 

Reviewer #1: Yes

Reviewer #2: Yes

4. Have the authors made all data underlying the findings in their manuscript fully available?

Reviewer #1: Yes

Reviewer #2: Yes

5. Is the manuscript presented in an intelligible fashion and written in standard English?

Reviewer #1: Yes

Reviewer #2: Yes

6. Review Comments to the Author

Reviewer #1: I am satisfied by the revisions. In particular, the limitations to causal inference are now clearly acknowledged.

Reviewer #2: Dear PONE Journal Team of editorials,

I am thankful for the chance given to me to review the manuscript titled “Association between Median Household Income, State Medicaid Expansion Status, and COVID-19 Outcomes Across US Counties”. The article has benefit for measuring the association between income, medical expansion across different countries in US even if the spelling for countries is ‘counties’. Here by the way what does across the US countries mean? Is US one or more country with fifty plus states? Anyways, the following are my comments;

Is incidence and mortality the only outcome? Why didn’t you focus on it mainly? Why not risk of being infected by COVID 19? Then the continuum of illness through the diseases process? Why not recovery? What are the complications? Try to address multiple whys?

Is the median income an established/factful US income before or during the era of the pandemic? Why median? Why not high income? Why not low income?

Did US have different Medicaid expansion for the high, medium and low wealth index?

How do you see the association between the income, Medicaid and COVID outcomes? If so let us take it as a sort of continuum of care, have you calculated the dropout rate per each level?

Use scientific methods of writing as per the standard. There are inconsistencies. Language, tense, sentences, ideas, paragraphs and the general manuscript needs to be revised?

Check the whole statistics again

Regards,

7. PLOS authors have the option to publish the peer review history of their article (what does this mean?). If published, this will include your full peer review and any attached files.

Reviewer #1: **Yes: **

Reviewer #2: **Yes: **

---

## [Author Response · Author response to Decision Letter 1]

29 Jun 2022

Author’s Reply to Reviewer’s Comments

Journal Requirements:

1/ Please ensure that your manuscript meets PLOS ONE's style requirements, including those for file naming. The PLOS ONE style templates can be found at

R/ We apologize for the not adhering by PLOS ONE's style requirements. We have altered the format of the manuscript to reflect the updated PLOS ONE's style requirements, including those for file naming.

2/ Thank you for stating the following financial disclosure: Yes, this project is supported by the Department of Veterans Affairs, Health Services Research Award IRP-20-003 (WW, project PI). Please state what role the funders took in the study. If the funders had no role, please state: "The funders had no role in study design, data collection and analysis, decision to publish, or preparation of the manuscript." If this statement is not correct you must amend it as needed. Please include this amended Role of Funder statement in your cover letter; we will change the online submission form on your behalf.

R/ We apologize for the omission regarding the role of the funders in the study. We have adjusted the financial disclosure statement to include a statement clarifying the role of the funders in the study.

3/ Thank you for stating the following in the Acknowledgments/ Funding Section of your manuscript: "This project is supported by the Department of Veterans Affairs, Health Services Research Award IRP-20-003 (WW, Project PI)". Please note that funding information should not appear in the Acknowledgments section or other areas of your manuscript. We will only publish funding information present in the Funding Statement section of the online submission form. Please remove any funding-related text from the manuscript and let us know how you would like to update your Funding Statement. Currently, your Funding Statement reads as follows: "Yes, this project is supported by the Department of Veterans Affairs, Health Services Research Award IRP-20-003 (WW, project PI)" Please include your amended statements within your cover letter; we will change the online submission form on your behalf.

R/ Thank you for bringing this to our attention. We have removed all information relating to the funding source in the Acknowledgments section and other areas of the manuscript. Additionally, we have updated the cover letter to include information regarding funding. We would like the online submission form to state the following: “This project is supported by the Department of Veterans Affairs, Health Services Research Award IRP-20-003 (WW, Project PI). The funders had no role in study design, data collection and analysis, decision to publish, or preparation of the manuscript.”

4/ We noted in your submission details that a portion of your manuscript may have been presented or published elsewhere. Please clarify whether this publication was peer-reviewed and formally published. If this work was previously peer-reviewed and published, in the cover letter please provide the reason that this work does not constitute dual publication and should be included in the current manuscript.

R/ While awaiting information a decision from PLOS ONE, the authors of the manuscript added portions of the manuscript to the preprint server ‘medRxiv’, an option endorsed by PLoS ONE. The preprints on medRxiv are initial reports, and as such, it is not peer-reviewed or considered formally published.

5/ Please amend your list of authors on the manuscript to ensure that each author is linked to an affiliation. Authors’ affiliations should reflect the institution where the work was done (if authors moved subsequently, you can also list the new affiliation stating “current affiliation:….” as necessary).

R/ We apologize for the oversight about the affiliations of the authors of the manuscript. We have amended the list of authors to reflect the authors affiliations during the time when the work was done.

6/ Please include captions for your Supporting Information files at the end of your manuscript, and update any in-text citations to match accordingly. Please see our Supporting Information guidelines for more information: http://journals.plos.org/plosone/s/supporting-information

R/ We have amended the in-text citations to match the PLOS ONE's style requirements. Also, the Supporting Information files are at the end of the manuscript.

8/ Thank you for including your ethics statement on the online submission form: "Providence Veterans Affairs Medical Center Institutional Review Board, formal waiver of approval due to non-Human subject research, project 1660196." To help ensure that the wording of your manuscript is suitable for publication, would you please also add this statement at the beginning of the Methods section of your manuscript file.

R/ We have revised the beginning of the Methods section of the manuscript to include the ethics statement from the Providence Veterans Affairs Medical Center Institutional Review Board.

R/ We have reviewed the reference list to ensure that it is complete and correct. During our review, we did not come across any cited papers that have since been retracted.

Author’s Responses to Reviewer's Questions:

Reviewer #1: I am satisfied by the revisions. In particular, the limitations to causal inference are now clearly acknowledged.

R/ Thank you for taking the time to provide insight into our work.

Reviewer #2: “Dear PONE Journal Team of editorials, I am thankful for the chance given to me to review the manuscript titled “Association between Median Household Income, State Medicaid Expansion Status, and COVID-19 Outcomes Across US Counties”. The article has benefit for measuring the association between income, medical expansion across different countries in US even if the spelling for countries is ‘counties’. Here by the way what does across the US countries mean? Is US one or more country with fifty plus states? Anyways, the following are my comments;”

R/ We appreciate your review of our work. After thorough review of the manuscript, we were unable to identify any occurrences of the use of ‘countries’. The analysis included 3142 United States counties (a political and administrative division within the 50 states) including the District of Columbia.

• “Is incidence and mortality the only outcome? Why didn’t you focus on it mainly? Why not risk of being infected by COVID 19? Then the continuum of illness through the diseases process? Why not recovery? What are the complications? Try to address multiple whys?”

R/ We appreciate your review of our work. In order to avoid over-extending the scope of paper and distract the reader, we have chosen to focus only on two main COVID outcomes, the COVID incidence and COVID mortality. As such, the incident rate ratio reported in our results would be considered an epidemiologically acceptable estimate of the relative risk of being infected with COVID and the mortality risk ratio would be an estimate of the worst complication of COVID, i.e. death. We did not address the recovery portion of COVID since it would be outside the scope of the current paper.

• “Is the median income an established/factful US income before or during the era of the pandemic? Why median? Why not high income? Why not low income?”

R/ Given that the unit of analysis being the county, the median income of the households within the county is commonly used as an estimate of the socioeconomic status for the population of the county. Since median income would vary by the socioeconomic status of the population of the county, it would allow us to compare the COVID incidence and mortality between “high income” counties versus “low income” counties. Together they answered our main study questions of “the impact of COVID-19 in the counties nationwide according to their socio-economic status and investigate whether the impact varies by counties of states with Medicaid expansion versus those without Medicaid expansion.”

• “Did US have different Medicaid expansion for the high, medium and low wealth index?”

R/ Excellent question. Medicaid provides a source of health insurance for low-income households and/or those with disability, among other factors. However, the income and qualification thresholds for enrollment into Medicaid differ by state, which depend on the state government’s policy. If a state has adopted the policy of Medicaid expansion, the requirements for Medicaid eligibility are less and at lower income thresholds, which makes it easier for people to enroll in health insurance versus states who did not adopt Medicaid expansion. 

• “How do you see the association between the income, Medicaid and COVID outcomes? If so let us take it as a sort of continuum of care, have you calculated the dropout rate per each level?”

R/ Based on the above, it is possible that COVID incidence or mortality may disproportionately affect counties of lower socioeconomic status from states that did not adopt Medicaid expansion because less people in that state would have access to health insurance. Less access to health insurance can lead to worse COVID complications, such as death, because the access to and the continuum of, care, are impaired. Therefore, we first compared the COVID incidence and mortality amongst counties of different socioeconomic status (based on median household income) to study this association. Mortality rates were calculated for each quartile of the median household income. We then looked at the impact of Medicaid expansion on this association by comparing the COVID mortality amongst counties of different socioeconomic status within states that adopted Medicaid expansion and those that did not. 

• “Use scientific methods of writing as per the standard. There are inconsistencies. Language, tense, sentences, ideas, paragraphs and the general manuscript needs to be revised?”

R/ We agree with the reviewer and have reviewed the manuscript to ensure the use of scientific methods of writing.

• “Check the whole statistics again”

R/ We appreciate your review of our work. We performed a thorough review of the statistics, we did not find any items in need of correction at this point.

We appreciate and thank the reviewer for his/her comments, to which we provided a point-to-point answer in the reply to reviewer’s document, all of which, made our paper a much improved one.

Tsikata Apenyo & Wen-Chih Wu

---

## [Decision Letter · Decision Letter 2]

15 Jul 2022

PONE-D-21-39999R2Association between Median Household Income, State Medicaid Expansion Status, and COVID-19 Outcomes Across US CountiesPLOS ONE

Dear Dr. Wu,

Thank you for submitting your manuscript to PLOS ONE. After careful consideration, we feel that it has merit but does not fully meet PLOS ONE’s publication criteria as it currently stands. Therefore, we invite you to submit a revised version of the manuscript that addresses the points raised during the review process. Please revise.

We look forward to receiving your revised manuscript.

Kind regards,

Academic Editor

PLOS ONE

Journal Requirements:

Reviewers' comments:

Reviewer's Responses to Questions

**Comments to the Author**

1. If the authors have adequately addressed your comments raised in a previous round of review and you feel that this manuscript is now acceptable for publication, you may indicate that here to bypass the “Comments to the Author” section, enter your conflict of interest statement in the “Confidential to Editor” section, and submit your "Accept" recommendation.

Reviewer #1: All comments have been addressed

Reviewer #3: (No Response)

2. Is the manuscript technically sound, and do the data support the conclusions?

Reviewer #1: (No Response)

Reviewer #3: Yes

3. Has the statistical analysis been performed appropriately and rigorously? 

Reviewer #1: (No Response)

Reviewer #3: Yes

4. Have the authors made all data underlying the findings in their manuscript fully available?

Reviewer #1: (No Response)

Reviewer #3: Yes

5. Is the manuscript presented in an intelligible fashion and written in standard English?

Reviewer #1: (No Response)

Reviewer #3: Yes

6. Review Comments to the Author

Reviewer #1: (No Response)

Reviewer #3: Thank you for providing the opportunity to review this revised manuscript. It appears that the authors have addressed many concerns from the previous round of review ( I was not part of the previous review).

The revised manuscript is generally well-presented and presents an interesting topic. Many of my comments are minor. My biggest concern is that the authors proceeded with the outcome of crude mortality and crude incidence rate. I know that US counties are very diverse in terms of age distribution and as such, the age-adjusted mortality rates and age-adjusted incidence rates would have been the better metrics. Please provide enough justification for not age-adjusting rates. Having said that, I have seen plenty of studies examining the crude rate, primarily because of the lack of data. If that is the case, please provide this information in the limitation section.

Key findings /Questions: household income should be replaced with county-level median HH income.

Abstract:

Objective: What you have as the objective in the abstract is not really an objective. That’s background information.

Methods: Use the easy-to-understand word format for the date given the global readership of the journal. Avoid star; either use the proper multiplication sign or have it in word.

Results: insert “rate” after COVID-19 incidence and COVID-19 mortality

Background: On your hypothesis, household income should be replaced with county-level median HH income.

Outcome: See my general comment on the top

Statistical Analysis:

For the sake of completeness, also mention that these income ranges are per annum.

“Using county population as the denominator in the model”: well there is no such thing as “denominator” in the model. Either you specify the rate (covid outcomes divided by denominator population) as the outcome, or most likely given the NB model, you specified the population as an offset. Make this clearer.

I see the value of having random intercepts for counties but not for the reason the authors suggest (i.e. to account for clustering, especially in conjunction with an unstructured covariance matrix). I would have used the robust standard errors clustered at the state level. But I am not holding this relatively small issue against the authors for their enormous effort.

“If the interaction proved to be significant”: note that you are not proving significance. I’d

rephrase this sentence.

Table 1 title” Again, avoid giving an impression as if the income is household level. Even if the software gives you P=0, this is theoretically not possible. Write something like P<0.0001.

7. PLOS authors have the option to publish the peer review history of their article (what does this mean?). If published, this will include your full peer review and any attached files.

Reviewer #1: No

Reviewer #3: No

---

## [Author Response · Author response to Decision Letter 2]

15 Jul 2022

Author’s Reply to Reviewer’s Comments

We appreciate and thank the reviewers for their comments, to which we provided a point-to-point answer below:

Reviewer #3: 

The revised manuscript is generally well-presented and presents an interesting topic. Many of my comments are minor. My biggest concern is that the authors proceeded with the outcome of crude mortality and crude incidence rate. I know that US counties are very diverse in terms of age distribution and as such, the age-adjusted mortality rates and age-adjusted incidence rates would have been the better metrics. Please provide enough justification for not age-adjusting rates. Having said that, I have seen plenty of studies examining the crude rate, primarily because of the lack of data. If that is the case, please provide this information in the limitation section.

R/ We agree with the reviewer that age-adjusted mortality rates and age-adjusted incidence rates would have been the better metrics. However, the data available does not have age-adjusted rates for which we have added this as a limitation in the discussion section (pg. 13, second to last paragraph). We also stated that we adjusted for age, gender, race and comorbidities of the county population to minimize the residual confounding.

Key findings /Questions: household income should be replaced with county-level median HH income.

R/ “household income” replaced with “county-level median household income” in Key findings /Questions

Abstract:

Objective: What you have as the objective in the abstract is not really an objective. That’s background information.

R/ We replace the sentence with “To study the relationship between county-level COVID-19 outcomes (incidence and mortality) and county-level median household income and status of Medicaid expansion of US counties”

Methods: Use the easy-to-understand word format for the date given the global readership of the journal. Avoid star; either use the proper multiplication sign or have it in word.

R/ We changed the date format to “January-20th-2021 through December-6th-2021”. We deleted the star and changed it to “Medicaid-expansion-times-income-quartile interaction”

Results: insert “rate” after COVID-19 incidence and COVID-19 mortality

R/ “rate” was inserted after COVID-19 incidence and COVID-19 mortality through out to be read as “incidence-rate” and “mortality-rate”.

Background: On your hypothesis, household income should be replaced with county-level median HH income.

R/ We replaced “household income” with “county-level median household income” in the hypothesis

Outcome: See my general comment on the top

R/ “rate” was inserted after COVID-19 incidence and COVID-19 mortality through out to be read as “incidence rate” and “mortality rate” or “incidence and mortality rates”.

Statistical Analysis:

For the sake of completeness, also mention that these income ranges are per annum.

“Using county population as the denominator in the model”: well there is no such thing as “denominator” in the model. Either you specify the rate (covid outcomes divided by denominator population) as the outcome, or most likely given the NB model, you specified the population as an offset. Make this clearer.

R/ “Median household income” was changed to “County-level median household income per annum” under Main Exposure Variables. We also made the same change to “county-level median household income per annum” under the statistical analysis. We also changed the word “denominator” to “offset”.

I see the value of having random intercepts for counties but not for the reason the authors suggest (i.e. to account for clustering, especially in conjunction with an unstructured covariance matrix). I would have used the robust standard errors clustered at the state level. But I am not holding this relatively small issue against the authors for their enormous effort.

R/ We thank the reviewer for their understanding.

“If the interaction proved to be significant”: note that you are not proving significance. I’d

rephrase this sentence.

R/ the sentence was rephrased to “If the interaction was significant…”

Table 1 title” Again, avoid giving an impression as if the income is household level. Even if the software gives you P=0, this is theoretically not possible. Write something like P<0.0001.

R/ We have changed to “County-Level Median Household Income” for titles and sub-titles of the tables. We changed the P values=0 to P<0.0001.

Thank you sincerely for the insightful comments which made our paper a much improved one.

Sincerely,

Tsikata Apenyo & Wen-Chih Wu

---

## [Decision Letter · Decision Letter 3]

21 Jul 2022

Association between Median Household Income, State Medicaid Expansion Status, and COVID-19 Outcomes Across US Counties

PONE-D-21-39999R3

Dear Dr. Wu,

We’re pleased to inform you that your manuscript has been judged scientifically suitable for publication and will be formally accepted for publication once it meets all outstanding technical requirements.

Kind regards,

Academic Editor

PLOS ONE

Additional Editor Comments (optional):

Reviewers' comments:

Reviewer's Responses to Questions

**Comments to the Author**

1. If the authors have adequately addressed your comments raised in a previous round of review and you feel that this manuscript is now acceptable for publication, you may indicate that here to bypass the “Comments to the Author” section, enter your conflict of interest statement in the “Confidential to Editor” section, and submit your "Accept" recommendation.

Reviewer #1: All comments have been addressed

Reviewer #3: All comments have been addressed

2. Is the manuscript technically sound, and do the data support the conclusions?

Reviewer #1: (No Response)

Reviewer #3: Yes

3. Has the statistical analysis been performed appropriately and rigorously? 

Reviewer #1: (No Response)

Reviewer #3: Yes

4. Have the authors made all data underlying the findings in their manuscript fully available?

Reviewer #1: (No Response)

Reviewer #3: (No Response)

5. Is the manuscript presented in an intelligible fashion and written in standard English?

Reviewer #1: (No Response)

Reviewer #3: Yes

6. Review Comments to the Author

Reviewer #1: (No Response)

Reviewer #3: Authors addressed all concerns I had. Thank you, and good luck with the next step of the publication.

7. PLOS authors have the option to publish the peer review history of their article (what does this mean?). If published, this will include your full peer review and any attached files.

Reviewer #1: No

Reviewer #3: No

---

## [Editor Report · Acceptance letter]

2 Aug 2022

PONE-D-21-39999R3 

Association between Median Household Income, State Medicaid Expansion Status, and COVID-19 Outcomes Across US Counties 

Dear Dr. Wu:

I'm pleased to inform you that your manuscript has been deemed suitable for publication in PLOS ONE. Congratulations! Your manuscript is now with our production department. 

Kind regards, 

on behalf of

Dr. Robert Jeenchen Chen 

Academic Editor

PLOS ONE